# Autoregressive Flow Matching for Motion Prediction

## Abstract

Motion prediction has been studied in different contexts with models trained on narrow distributions and applied to downstream tasks in human motion prediction and robotics. Simultaneously, recent efforts in scaling video prediction have demonstrated impressive visual realism, yet they struggle to accurately model complex motions despite massive scale. Inspired by the scaling of video generation, we develop autoregressive flow matching (ARFM), a new method for probabilistic modeling of sequential continuous data and train it on diverse video datasets to generate future point track locations over long horizons. To evaluate our model, we develop benchmarks for evaluating the ability of motion prediction models to predict human and robot motion. Our model is able to predict complex motions, and we demonstrate that conditioning robot action prediction and human motion prediction on predicted future tracks can significantly improve downstream task performance.

## 1 Introduction

Consider the video frame in the top left of Figure 1 where we see a person standing above a slackline. One plausible future motion would be for the man to fall down then bounce back up on the slackline, or miss and fall to the ground. However, it would not be plausible for the person to fall through the ground or rise above the frame. While humans can easily understand plausible future motion given just a single video frame, developing a method to probabilistically model future motion is challenging as it requires understanding agent behavior and physics. In this paper, we aim to predict future motion with vision-language conditioning on a wide range of video domains.

Due to the challenging nature of motion prediction, prior works focus on narrow domains such as human motion Walker et al. (2015); Luo et al. (2017); Chen et al. (2023), self-driving Zhang et al. (2024); Mercat et al. (2020); Nayakanti et al. (2023), and robotics Bharadhwaj et al. (2024); Wen et al. (2023); Xu et al. (2024); Yuan et al. (2024). While these are successful in their respective domains, they require domain-specific knowledge and lack generality. On the other hand, in recent advances in video generation Veo-Team et al. (2024); OpenAI (2024), large-scale training on internet datasets has unlocked the ability to generate visually plausible and diverse videos. Yet, these models often struggle to accurately model motion and realistic physics, and remain prohibitively expensive due to pixel-level synthesis. We take inspiration from both lines of work and explore generalized motion prediction, a formulation that combines the scalability and flexibility of large-scale video models with the efficiency and structure of motion-centric prediction.

Unlike prior motion prediction works which focus on a specific domain and video prediction works which focus on visual realism, our proposed formulation aims to predict future motion for a wide range of videos. The task of predicting track motion presents a unique challenge of modeling a multivariate causal time series with vision and language conditioning. Furthermore, since future motion has irreducible uncertainty, there are many future plausible motions when provided few conditioning frames, we must model the future point tracks probabilistically. Lastly, at inference time, we require the ability to efficiently roll out predictions and quickly update them in response to newly provided frames while remaining temporally consistent.

To this end, we propose autoregressive flow matching, a new method that combines the strengths of autoregressive modeling with flow matching prediction to probabilistically model future motion. We train our model using pseudo-labels from an off-the-shelf point tracker, which enables our method to easily scale to a

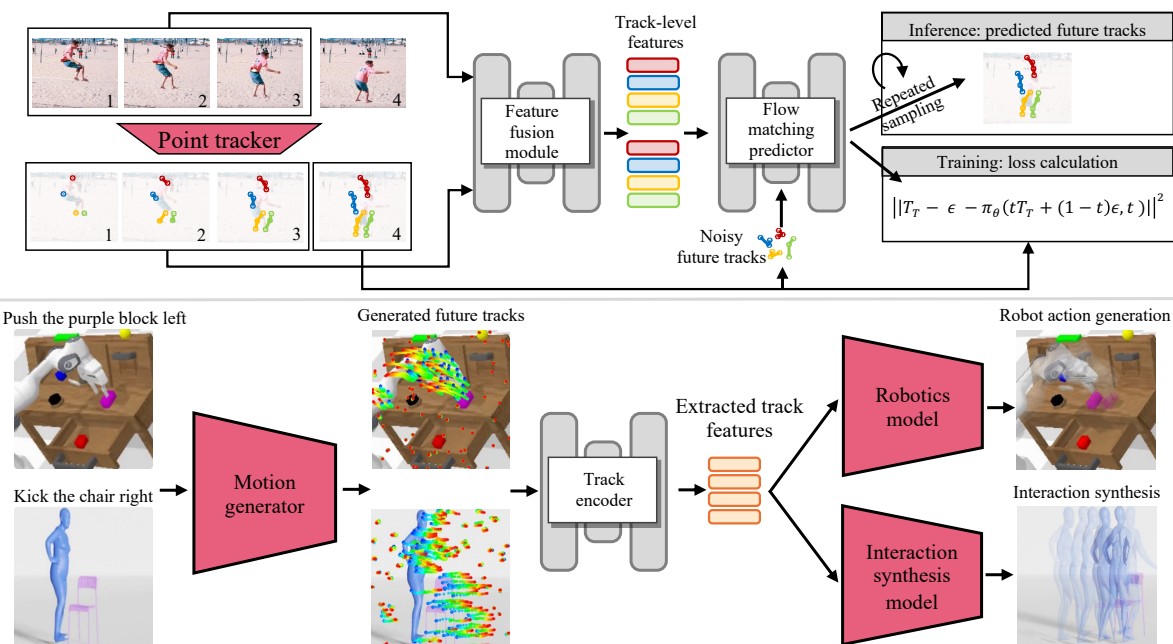

Figure 1: **Generalized motion prediction via autoregressive flow matching**. To train our method (top), we extract pseudo label tracks using an off-the-shelf point tracker, then train an autoregressive flow matching model to predict tracks. To apply our method to downstream tasks (bottom), we generate future point tracks conditioned on provided video frames and an optional language prompt. Then, we extract features from these tracks which are used by a task specific model to accomplish the task.

large number of diverse video datasets. When trained on these datasets, our motion predictor is capable of producing accurate predictions conditioned on as few as a single frame. Furthermore, we demonstrate the utility of our model by using predicted tracks from our model to condition robotic action prediction and human interaction synthesis, which results in significant improvements in task performance. Our main contributions are:

1. Autoregressive flow matching (ARFM), a new method for efficient probabilistic modeling of sequential continuous data.

2. Methods for predicting future point tracks, updating a constant track horizon in an online fashion, predicting query point information, and encoding point tracks to features, which improve task performance in robotics and human interaction synthesis.

3. Track motion prediction benchmarks and training datasets developed using a pseudo-labeling approach to train and evaluate future point track prediction.

## 2 Related Work

**Optical flow and point tracking.** Optical flow is a classic computer vision problem which involves estimating pixel motion between two video frames. A related task is that of point tracking which is the task of predicting point pixel motion and visibility over many frames. Recently, various works (Teed & Deng, 2020; Harley et al., 2022; Zheng et al., 2023) have improved performance on these two problems through the use of deep learning. Similarly, other works (Wang et al., 2023; Doersch et al., 2024) have improved the tracking of points over longer durations and through occlusions. With these recent advancements, we are able to use existing point trackers to pseudo-label videos for training our track prediction model.

**Video generation**   Video generation (Wu et al., 2021; Gupta et al., 2022; Höppe et al., 2022; Finn et al., 2016) is closely related to our proposed motion generation task and involves predicting future video frames given past video frames. Previously, video generation has been used to provide a dynamics model for robotics trajectory planning Du et al. (2023); Finn et al. (2016), and recent advancements in video synthesis allow for the generation of visually plausible videos from natural language instructions. While these video generation models create visually appealing videos, they still struggle to accurately model complex motion and are incredibly expensive at inference time due to their massive scale. In comparison, our proposed problem seeks to only model the motion aspects of video and can run efficiently by predicting only sparse point tracks.

**Motion prediction.**   Prior works have explored the problem of motion prediction in different contexts. In robotics (Bharadhwaj et al., 2024; Xu et al., 2024; Wen et al., 2023; Yuan et al., 2024), recent works have learned conditional motion prediction methods for the purpose of predicting plausible trajectories or learning affordances (Bahl et al., 2023). Other works (Luo et al., 2017; Walker et al., 2015; Cao et al., 2020; Shi et al., 2024) have trained models to predict short-term object motion on human centric datasets to learn priors for human action recognition and human motion prediction. Many works (Zhang et al., 2024; Mercat et al., 2020; Nayakanti et al., 2023) have also studied motion prediction in the context of autonomous driving in order to predict the future occupancy locations of cars in lidar and graph modalities. Compared to these prior works, we perform motion prediction in a generalized fashion with a single foundation model trained on a diverse set of videos enabling zero-shot predictions.

## 3   Approach

In this section, we present our approach to the various sub-problems of generalized motion prediction. We first discuss our pseudo-labeling data pipeline which enables us to train our method completely on unlabeled video sources. Next, we provide an overview of autoregressive flow matching and our architecture for motion prediction. Finally, we discuss how we perform online track prediction updates and feature encoding for downstream applications.

### 3.1   Data pipeline

We assume access to a large unlabeled video database, which in practice can be collected relatively easily through merging existing video datasets and web videos. For each video $V \in \mathbb{R}^{L \times H \times W \times 3}$ where $L$ is the number of frames in the video, $H$ is the height, and $W$ is the width, we use the BootsTAP Doersch et al. (2024) point tracker for its high accuracy while also not conditioning on other tracked points. We perform a single large inference pass to get tracks for a large number of query points paired with each video resulting in predicted tracks $T \in \mathbb{R}^{M \times L \times 2}$ where $M$ is the total number of points per video. Last, along with each video we have additional conditioning values such as descriptions of natural language and frame rate.

**Track sub-sampling**   While a large number of initial tracks are collected, during training we perform random subsampling. To subsample these tracks, we first perform a minimum visibility ratio filtering where only tracks which are visible for greater than half of the future frames are selected for prediction because we only supervise the prediction on frames where a track is marked as visible. Following this, we perform movement-weighted sampling. For a varying temperature $\tau$, the sampling probability for each point is equal to the temperature scaled softmax of the average squared movement between each frame. To construct input tracks and target tracks to train our method, we slice the tracks such that the input tracks are every track except for the last video timestep $T_I = T[:,\text{:-1}]$ and the prediction targets $T_T = T[:,1:] - T[:,\text{:-1}]$ are the relative shift for the next track. $T_I, T_T \in \mathbb{R}^{N \times L-1 \times 2}$ where $N$ is the number of tracks after subsampling.

### 3.2   Future track prediction model

**Autoregressive flow matching**   Track prediction presents a unique challenge of predicting plausible future trajectories in the presence of irreducible noise efficiently and in an online fashion. To address these challenges, we present autoregressive flow matching, a new modeling method that separates conditioning on previous input values from the denoising procedure. Whereas previous works Chen et al. (2025); Wu et al.

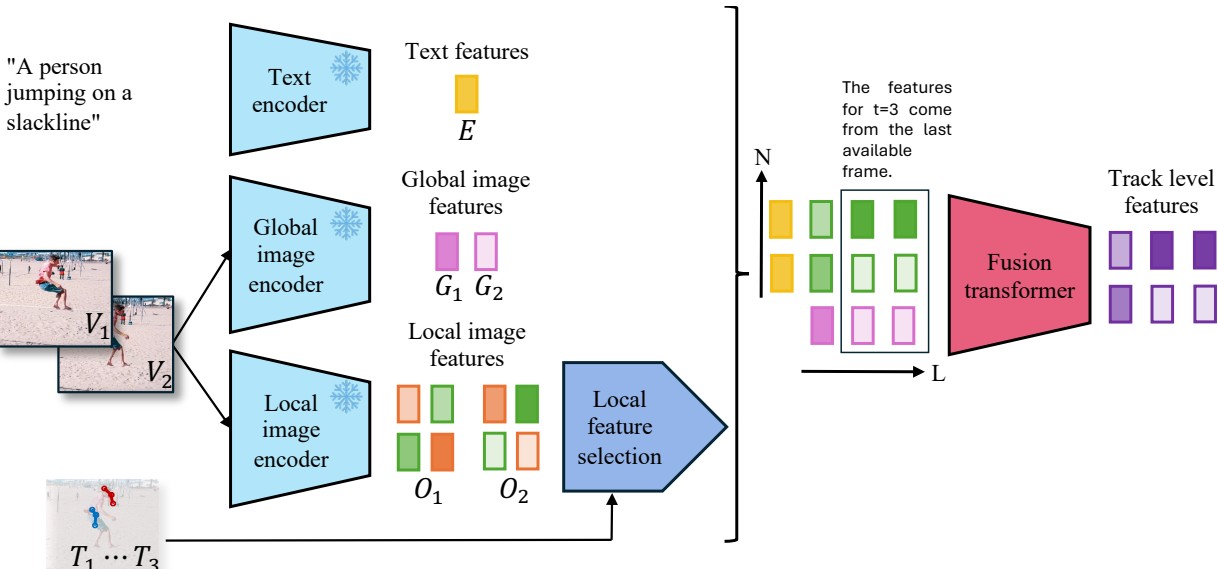

Figure 2: **Overview of feature fusion module**. Our feature fusion module which takes video frames, past point tracks, and an optional natural language caption. These multi-modal inputs are aggregated into a single feature vector for each future track prediction location.

(2024) concatenate a noise vector along with input values, we instead predict outputs autoregressively in two separate stages. First, a feature fusion module aggregates past input information into a feature vector for the current timestep. Second, a flow matching predictor observing only the feature vectors from the current timestep denoises targets for only the current timestep. By splitting the past conditioning from the flow matching prediction into separate modules, we only need to perform sampling forward passes of the final flow matching predictor which is relatively lightweight compared to the feature fusion module. Additionally, we avoid a train vs. test time mismatch or having to store past noise trajectories. We expand on this issue in the appendix. While autoregressive flow matching can be used to model any sequential continuous data, below we describe a feature fusion module and flow matching predictor design tailored for the track prediction task.

**Feature fusion module** An overview of our feature fusion module is shown in Figure 2. Prior to feature fusion, three foundation models are used to extract relevant features from the various inputs. During training, a random starting point is selected for both video frames and tracks, and a random number $L_c$ of initial frames between 1 and $L-1$ are selected to be used as video conditioning. A high resolution image model is used to extract hierarchical feature maps of varying resolutions $O \in \mathbb{R}^{L_c \times H//S_i \times W//S_i \times C_i}$ where $S_i$ is the downsample ratio for the $i$-th feature map and $C_i$ is the number of feature channels for the $i$-th feature map. Next, for each feature map resolution level $O_i$, we use input tracks $T_I$ to index the feature map spatio-temporally with nearest neighbor interpolation. Then, we concatenate the selected features for each feature map level channel-wise to obtain local features $O^{T_I} \in \mathbb{R}^{L-1 \times N \times \sum_i C_i}$. Since the number of provided frames is always less than the input track length, for input tracks on non-visible frames, we select from the last feature map video timestep. To differentiate between visible and non-visible frames we sum an additional categorical embedding indicating whether the feature is from a visible frame. Following this, we also extract low resolution image features $G \in \mathbb{R}^{L_c \times (H//S_l)*(W//S_l) \times C_l}$ which are spatially concatenated for cross attention. We use SAMv2 Ravi et al. (2024) for the local feature extractor to encode high frequency image features and positional information and the image branch of CLIP Radford et al. (2021) for global image features which allows for each query point to attend to the full image information regardless of what other query points exist in the sample.

Lastly, language embeddings are extracted using a text encoder which is the text branch of CLIP. These embeddings are $E \in \mathbb{R}^{L_e \times C_e}$ where $L_e$ is the tokenized sequence length and $C_e$ is the number of hidden channels. We expand these language embeddings by repeating in the spatial axis then prepend along the temporal axis

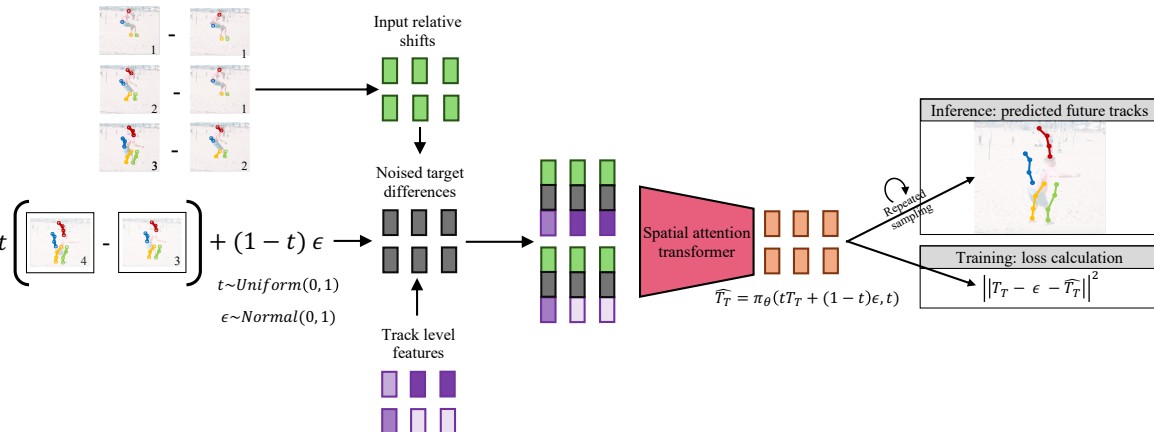

Figure 3: **Overview of training flow matching predictor**. Using each of the track level features and relative shift inputs, the model attends spatially to denoise temporal track differences which are used during inference time for sampling.

along with the other features to form the final input features $H \in \mathbb{R}^{L_t + L_c \times N + (H//S_l)*(W//S_l) \times C_h}$. Throughout the process, channel mismatches when concatenating features are resolved by using linear projections to a common dimensionality.

These input features are then processed using a spatiotemporal transformer. Instead of using full spatiotemporal attention which would greatly increase the compute cost due to the quadratic complexity of attention, we alternate between layers of causal attention along the time axis and bidirectional attention along the spatial axis. We then select the track level features of the output of this module to obtain the track level features for flow matching prediction. Because the features are causally masked, we can maintain a key-value cache at inference time to efficiently generate the features for the next timestep when encoding a new frame. Since the attention along the time axis of our fusion transformer is causally masked, during inference time we can efficiently generate the features for the next video timestep using a key value cache.

**Flow matching predictor**  An overview of our flow matching predictor module is provided in Figure 3. Flow matching Lipman et al. (2022) is a method to learn ordinary neural differential equations commonly used for generative modeling. For our purposes, we perform conditional prediction of point track differences as follows. First, we sample a denoising timestep $t \sim Uniform(0,1)$ and Gaussian noise with the same shape as target track shifts $\epsilon \sim \mathcal{N}(0,I)$ with $\epsilon \in \mathbb{R}^{N \times L-1 \times 2}$.

We create a noised target as a linear combination of the true target shifts and sampled noise $tT_T + (1-t)\epsilon$. These noised target shifts are then concatenated channel-wise as the input to the flow matching predictor. Additionally, we concatenate the relative shift from between the current point and previous point as additional conditioning constructed as $T_C = (0^{N \times 1 \times 2}, T_I[:, 1:] - T_I[:, :-1])$. This is equivalent to the target shifts for the previous video timestep at each input video timestep, but we also pad the first video timestep with just a zero value as there is no prior track to calculate a shift.

To ensure that any uncertainty is modeled jointly between two points, we must condition predictions on other point tracks. For example, multiple query points on the same rigid object should move jointly in future predictions which can be only achieved through joint prediction. We construct flow targets for the flow matching predictor as the difference between the sampled noise and target relative shifts. When fitting parameters $\theta$ for the combined feature fusion module and flow matching predictor $\pi_\theta$, the following objective is optimized during training

$$\underset{\theta}{\operatorname{argmin}} \; \mathbb{E}[||T_T - \epsilon - \pi_\theta(tT_T + (1-t)\epsilon, t, V, T_I)||^2]. \tag{1}$$

The entire network is optimized end-to-end using stochastic gradient descent. During inference time, we denoise the track at each video timestep fully using the current timestep track level features. Then, the fully denoised tracks are used as conditioning to generate the features for the next timestep. We autoregressively

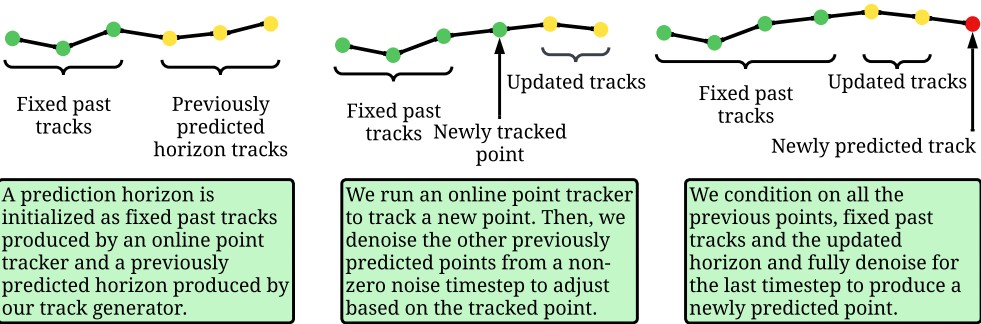

Figure 4: **Horizon update overview.** To efficiently update a predicted horizon, we first obtain a newly tracked point using an online point tracker. Then, we apply noise to previously predicted tracks followed by denoising from an intermediate timestep to update the tracks while retaining continuity. Finally, we fully denoise a new point at the final timestep.

continue this process by continually denoising timesteps and using denoised predictions to predict future outputs until we generate a prediction horizon of the desired length. Since the flow matching predictor only attends spatially, we are able to perform efficient inference by only running the flow predictor on the video timestep we are predicting.

## 3.3 Query information predictor

Since during training we upsample queries with higher movements, we create a non-trivial point sampling distribution. Therefore, for tasks where there are no provided query points, we must replicate a similar point sampling distribution at inference time for the most accurate track prediction results. To do so, we train a lightweight query information predictor which aims to predict the expected future movement magnitude for a large number of random query points. These expected future movement magnitudes are then used for weighted sampling as described in Section 3.1. This model creates input features by selecting features from the high resolution image feature extraction model, then predicts the expected movement using squared error regression.

## 3.4 Updating horizon prediction

Many tasks, such as robotic action prediction, require predicting a constant point horizon which is quickly updated as new frames are received and processed. While the naive solution is to simply generate a new prediction of the horizon length $h$ whenever a new frame is received, in practice this method is inefficient and leads to high variance between predictions when there is substantial uncertainty. Instead, we develop a consistent and accurate method for achieving updating horizon prediction by taking advantage of the fact that we can denoise predictions from an intermediate video timestep. An overview of this process is shown in Figure 4.

For applications requiring an updated horizon, we have the fixed past frame tracks $T_f \in \mathbb{R}^{N \times L_f \times 2}$ initialized as only query point positions, and a previously predicted horizon $T_h \in \mathbb{R}^{N \times L_h \times 2}$ as state. When a new frame is encoded, we run an online point tracker on the new frame to efficiently compute the fixed past point track for video timestep $L_f + 1$. The computed new point position then replaces the track at the 0-th position in the past generated horizon.

Since these two points may differ by some amount, we now perform an update on the horizon tracks from $L_f + 2$ to $L_h$. To do so, we perform the autoregressive flow matching sampling procedure from a denoising timestep $t_e \in [0, 1]$. Using an intermediate denoising timestep allows us to control the tradeoff between keeping a past prediction and making edits. After editing the past horizon, we perform a full denoising from $t = 0$ to obtain the new track position at video timestep $L_f + L_h + 1$ which is then concatenated to the old horizon in order to form the new full horizon.

### 3.5 Track conditioning applications

**Track feature extraction**   To condition on predicted track values in downstream tasks, we require a track encoding architecture that can efficiently extract features from raw predicted track values. For a set of predicted tracks $\hat{T} \in \mathbb{R}^{N \times L \times 2}$, we process the tracks into tokens as follows. First, we take the initial track location $T[:, 0]$ to be used as a "positional" embedding which is encoded using 2d fourier features of shape $\mathbb{R}^{N \times C}$. Then, for all the tracks, we compute the track shift at each video timestep as $T[:, 1:] - T[:, :-1]$ which are then flattened along the last two axes to form track shift features of shape $\mathbb{R}^{N \times 2(L-1)}$. The positional embeddings and track shift features are projected to a common channel dimension then summed together to form feature vectors. These feature vectors are processed with a ViT-like Dosovitskiy et al. (2020) bidirectional transformer architecture with a pooling [CLS] token for information aggregation.

**Track conditioning.**   To use tracks in downstream tasks such as robotics and interaction synthesis, we need to train the downstream model to take predicted tracks as an input. To do so, we first predicted ground truth future tracks which can be obtained simply with a point tracker model as we already have access to the full videos. Then, we feed the ground truth future point tracks into the aforementioned track feature extractor which produces a conditioning vector that is fed into the downstream application model. We opt to use ground-truth future tracks during training so that a single trained application model can be used with various track prediction models without re-training, and predicted tracks which more closely match ground truth tracks at inference time will lead to better results.

## 4 Experiments

In this section, we evaluate our proposed point prediction model on the human motion and robotics domains.

### 4.1 Pre-training data

We train on the concatenation of Kinetics-400 (Kay et al., 2017), Motion-X (Lin et al., 2023) and robotics datasets from Open-X-Embodiment (O'Neill et al., 2024), including Language Table (Lynch et al., 2023), BridgeDataV2 (Walke et al., 2023), Maniskill (Mu et al., 2021), and MimicGen (Mandlekar et al., 2023). We choose to exclude the downstream evaluation datasets from our zero-shot pre-training evaluation in order to effectively test the generalization of our system to entirely unseen video datasets. This setup is especially challenging in the CALVIN (Mees et al., 2022b) benchmark where the simulator rendering differs from any of the training datasets.

### 4.2 Track prediction evaluation

To test the track prediction capabilities of our model, we perform evaluation on UCF-101 (Soomro et al., 2012) and the CALVIN robotics benchmark, following their exact setting and metrics. In addition to evaluating in a zero-shot setting, we also show results with a fine-tuned model on each downstream dataset.

To evaluate the quality of track predictions, we develop a metric based on point-tracker threshold based metrics. Specifically, we measure the ratio of predicted future point tracks within a certain euclidean distance from the ground truth point track. Evaluation is done using a single conditioning frame followed by prediction of 50 frames into the future or approximately 2 seconds for the frame rates in these video datasets. Due to the nature of the track prediction task which has substantial irreducible uncertainty, we sample each model five times for a data sample and take the best result when measuring $< \delta^x$ metrics. When measuring average displacement error (ADE), we simply take the average displacement error over all samples.

Since many video datasets have samples that are still frames and have almost no movement, we seek to filter for evaluation samples that are most informative for the motion prediction task. To filter samples with at least some motion, we calculate the euclidean distance between each pair of adjacent video timesteps, summed across the entire video sequence. Then, we take the average movements from the top 100 out of 1000 point tracks and only select evaluation samples with movements greater than 50. We will release our evaluation datasets along with the rest of our implementation.

Table 1: Comparison of methods on UCF-101 and CALVIN ABC → D by $< \delta^x$, measuring the ratio of points within $x$ pixels of the target and $ADE$ measuring the average displacement error between predicted and ground truth tracks at 256×256 resolution.

| Method | UCF-101 | | | | | | | CALVIN ABC → D | | | | | | |
|---|---|---|---|---|---|---|---|---|---|---|---|---|---|---|
| | $< \delta^4$ | $< \delta^8$ | $< \delta^{16}$ | $< \delta^{32}$ | $< \delta^{64}$ | $< \delta^x_{avg}$ | $ADE$ | $< \delta^4$ | $< \delta^8$ | $< \delta^{16}$ | $< \delta^{32}$ | $< \delta^{64}$ | $< \delta^x_{avg}$ | $ADE$ |
| No movement | **0.259** | 0.381 | 0.521 | 0.715 | 0.889 | 0.553 | 25.9 | 0.367 | 0.391 | 0.442 | 0.533 | 0.719 | 0.490 | 37.2 |
| Chronos Absolute Ansari et al. (2024) | 0.143 | 0.240 | 0.380 | 0.590 | 0.812 | 0.430 | 98.5 | 0.172 | 0.230 | 0.317 | 0.451 | 0.669 | 0.368 | 84.9 |
| Chronos Relative Ansari et al. (2024) | 0.209 | 0.341 | 0.498 | 0.697 | 0.880 | 0.523 | 54.5 | 0.235 | 0.339 | 0.429 | 0.529 | 0.717 | 0.448 | 30.2 |
| LARP Wang et al. (2024) | 0.151 | 0.262 | 0.414 | 0.652 | 0.880 | 0.468 | 36.1 | – | – | – | – | – | – | – |
| iVideoGPT Wu et al. (2025) | – | – | – | – | – | – | – | 0.359 | 0.434 | 0.501 | 0.600 | 0.772 | 0.530 | 34.1 |
| Cosmos Video2World Agarwal et al. (2025) | 0.209 | 0.351 | 0.514 | 0.721 | 0.893 | 0.538 | 30.1 | 0.254 | 0.336 | 0.418 | 0.529 | 0.730 | 0.453 | 39.6 |
| ARFM (ours) zero-shot | 0.237 | 0.386 | **0.581** | **0.788** | **0.935** | **0.586** | 24.5 | 0.365 | 0.405 | 0.486 | 0.659 | 0.909 | 0.565 | 33.6 |
| ARFM (ours) fine-tuned | 0.252 | **0.389** | 0.548 | 0.735 | 0.902 | 0.565 | **19.2** | **0.437** | **0.578** | **0.800** | **0.944** | **0.985** | **0.749** | **19.1** |

Table 2: **Interaction synthesis** on the FullBodyManipulation dataset Li et al. (2023). When conditioning on future track generations from our model, interaction synthesis quality significantly improves.

| Method | Condition Matching | | | Human Motion | | | | Interaction | | | | | GT Difference | | | |
|---|---|---|---|---|---|---|---|---|---|---|---|---|---|---|---|---|
| | $T_s \downarrow$ | $T_e \downarrow$ | $T_{xy} \downarrow$ | $H_{feet} \downarrow$ | FS$\downarrow$ | $R_{prec} \uparrow$ | $FID \downarrow$ | $C_{prec} \uparrow$ | $C_{rec} \uparrow$ | $C_{F_1} \uparrow$ | $C_\% \uparrow$ | $P_{hand} \downarrow$ | MPJPE$\downarrow$ | $T_{root} \downarrow$ | $T_{obj} \downarrow$ | $O_{obj} \downarrow$ |
| Interdiff Xu et al. (2023) | 0.00 | 158.84 | 72.72 | **0.90** | 0.42 | 0.08 | 208.0 | 0.63 | 0.28 | 0.33 | 0.27 | 0.55 | 25.91 | 63.44 | 88.35 | 1.65 |
| MDM Tevet et al. (2022) | 5.18 | 33.07 | 19.42 | 6.72 | 0.48 | 0.51 | 6.16 | 0.72 | 0.47 | 0.53 | 0.43 | 0.66 | 17.86 | 34.16 | 24.46 | 1.85 |
| Lin-OMOMO Li et al. (2023) | 0.00 | 0.00 | 0.00 | 7.21 | 0.41 | 0.29 | 15.33 | 0.68 | 0.56 | 0.57 | 0.54 | **0.51** | 21.73 | 36.62 | 17.12 | 1.21 |
| Pred-OMOMO Li et al. (2023) | 2.39 | 8.03 | 4.15 | 7.08 | 0.40 | 0.54 | 4.19 | 0.73 | 0.66 | 0.66 | **0.62** | 0.58 | 18.66 | 28.39 | 16.36 | 1.05 |
| GT-OMOMO Li et al. (2023) | 0.00 | 0.00 | 0.00 | 7.10 | 0.41 | 0.48 | 5.69 | 0.77 | 0.66 | 0.67 | 0.59 | 0.55 | 15.82 | 24.75 | 0.00 | 0.00 |
| CHOIS | 1.71 | **6.31** | 2.87 | 4.20 | **0.35** | 0.64 | 0.69 | **0.80** | 0.64 | 0.67 | 0.54 | 0.59 | **15.30** | 24.43 | 12.53 | 0.99 |
| CHOIS w/ ARFM (ours) zero-shot | **1.55** | 6.58 | **2.64** | **4.08** | **0.35** | 0.69 | 0.64 | **0.80** | 0.68 | 0.71 | 0.60 | 0.65 | 15.41 | 22.71 | 11.95 | 0.94 |
| CHOIS w/ ARFM (ours) fine-tuned | **1.55** | 6.58 | **2.64** | **4.08** | **0.35** | **0.72** | **0.63** | **0.80** | **0.70** | **0.72** | 0.60 | 0.65 | 15.41 | **22.89** | **11.90** | **0.94** |
| CHOIS w/ ground truth tracks | 1.52 | 6.54 | 2.64 | 4.10 | 0.35 | 0.70 | 0.61 | 0.81 | 0.72 | 0.73 | 0.62 | 0.64 | 15.22 | 22.88 | 11.74 | 0.95 |

**Track prediction baselines.** We create a few baselines for track prediction in our proposed setting. First, a no-movement baseline that only predicts the point track location at the last provided frame for the entire prediction horizon. Next, we consider treating the problem as a time series forecasting task to establish a prediction baseline without any visual inputs. For this, we use the Chronos (Ansari et al., 2024) time series foundation model to forecast future track locations given past track locations. For this model, we consider two formulations which are "absolute" prediction where the conditioning and prediction is performed given the raw point track values and "relative" where conditioning and prediction is the difference between each track. Lastly, we consider video generators LARP (Wang et al., 2024) for human motion, iVideoGPT (Wu et al., 2025) for robotics, and Cosmos Video2World (Agarwal et al., 2025) for both human motion and robotics. We apply the same point tracking model used for pseudo-labeling on the generated video frames of these models to create future point forecasts.

**Human motion track prediction.** The human motion point prediction results are shown in the left half of Table 1. We find that our method outperforms all of the baselines however, the simple no movement baseline outperforms all other baselines at lower track thresholds. When examining the predicted tracks and videos for the other baselines, we find that they struggle to predict still tracks on background points due to low temporal consistency and even when upsampling high movement tracks, background tracks still account for a significant portion of tracks.

**Robotics track prediction.** For downstream robotics evaluation, we benchmark on the CALVIN ABC → D split. Our results are shown in the right half of Table 1. We find that our proposed method significantly outperforms all baselines and that the fine-tuned model performs better than the zero-shot model. Most likely this is due to the fine-tuned model adapting to the simulator rendering which does not appear at all in the training data.

### 4.3 Interaction synthesis evaluation

Table 3: **CALVIN Benchmark Robot evaluation results**. We benchmark on the ABC → D and 10% ABCD → D data settings. When conditioning on generated future tracks from our model, task success rate significantly improves.

| Method | Split | Tasks completed in a row | | | | | |
|---|---|---|---|---|---|---|---|
| | | 1 | 2 | 3 | 4 | 5 | Avg. Len. |
| RT-1 Brohan et al. (2022) | ABC→D | 0.533 | 0.222 | 0.094 | 0.038 | 0.013 | 0.90 |
| HULC Mees et al. (2022a) | ABC→D | 0.418 | 0.165 | 0.057 | 0.019 | 0.011 | 0.67 |
| MT-R3M Nair et al. (2022) | ABC→D | 0.529 | 0.234 | 0.105 | 0.043 | 0.018 | 0.93 |
| GR-1 Wu et al. (2023) | ABC→D | 0.854 | 0.712 | 0.596 | 0.497 | 0.401 | 3.06 |
| GR-1 repro. | ABC→D | 0.858 | 0.732 | 0.660 | 0.574 | 0.494 | 3.31 |
| GR-1 + ARFM (ours) zero-shot | ABC→D | 0.917 | 0.816 | 0.723 | 0.641 | 0.561 | 3.66 |
| GR-1 + ARFM (ours) fine-tuned | ABC→D | **0.928** | **0.832** | **0.744** | **0.652** | **0.560** | **3.74** |
| RT-1 Brohan et al. (2022) | 10% data | 0.249 | 0.069 | 0.015 | 0.006 | 0.000 | 0.34 |
| HULC Mees et al. (2022a) | 10% data | 0.668 | 0.295 | 0.103 | 0.032 | 0.013 | 1.11 |
| MT-R3M Nair et al. (2022) | 10% data | 0.408 | 0.146 | 0.043 | 0.014 | 0.002 | 0.61 |
| GR-1 Wu et al. (2023) | 10% data | 0.778 | 0.533 | 0.332 | 0.218 | 0.139 | 2.00 |
| GR-1 repro. | 10% data | 0.768 | 0.555 | 0.380 | 0.261 | 0.176 | 2.14 |
| GR-1 + ARFM (ours) zero-shot | 10% data | 0.838 | 0.592 | **0.407** | 0.293 | 0.209 | 2.32 |
| GR-1 + ARFM (ours) fine-tuned | 10% data | **0.841** | **0.608** | 0.399 | **0.295** | **0.214** | **2.35** |

To evaluate how well our track prediction model can guide human object interaction synthesis, we condition an interaction synthesis model (Li et al., 2024) on tracks predicted by ARFM. The base human motion prediction model is CHOIS (Li et al., 2024) trained on FullBodyManipulation (Li et al., 2023), and we train the model to take inputs from either our pretrained zero-shot motion predictor, or a variant fine-tuned with track pseudo labels on FullBodyManipulation. We follow the single window setting of CHOIS, and our method simply adds conditioning on future predicted tracks without any other modifications. We report the results of this experiment in Table 2 and use the exact same setup and metrics. We find that the additional future track conditioning significantly improves human motion prediction results with the fine-tuned model providing slightly better results. In addition, we report results using ground truth tracks as an oracle which achieves the best results demonstrating improved future tracks also improve interaction synthesis quality.

## 4.4 Robotics evaluation

To evaluate the performance of our track prediction model at guiding robotics trajectory prediction, we condition a robotics model on tracks predicted by either our zero-shot pre-trained motion predictor or a variant fine-tuned with track pseudo labels on the CALVIN benchmark. For our base robotics model, we use the GR-1 (Wu et al., 2023) robotics model to demonstrate that track prediction can provide substantial performance improvement on top of an already strong robotics model. We evaluate on the more challenging ABC → D setting where no samples of the D environment are provided during finetuning of the base robotics model. In addition, we benchmark in a 10% data setting which uses the ABCD → D split, but only 10% of the total training data is used. Since there is no official public implementation for fine-tuning GR-1, we produce the results using the fine-tuning results from the publicly released pre-trained checkpoint and report our reproduced results. Aside from the baseline GR-1 (Wu et al., 2023) model with no track conditioning, we also compare with RT-1 (Brohan et al., 2022), HULC (Mees et al., 2022a), and a multi-task R3M (Nair et al., 2022). We present the results in Table 3. We find that when conditioning on predicted tracks, there is a significant improvement in the success rate across all chained in a row task lengths for both settings. Similar to our track prediction evaluations, we also find the fine-tuned model outperforms the zero-shot model.

## 5 Conclusion

In this paper, we introduce a method for generalized motion prediction that trains a singular model that can provide accurate future track predictions in a zero-shot setting. To accomplish this, we introduce autoregressive flow matching, a new method for modeling continuous sequential data as well as methods for

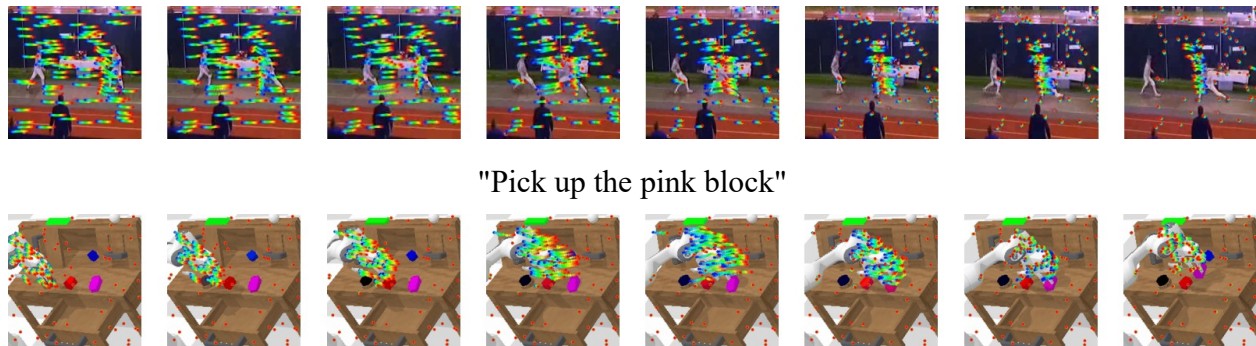

Figure 5: **Motion generation visualizations.** In the top row, we show a visualization on the evaluation set of UCF-101. We find our model can correctly predict the right fencer lunging toward the left one as well as predict the proper camera motion of the background moving right as the camera pans left to follow the motion of the fencers. In the bottom row, we show a visualization on the CALVIN robotics benchmark. Here, we see the model can properly guide the robot arm to the correctly colored block to complete the task. Furthermore, through our updating horizon prediction, the model is capable of differentiating between the block colors and correctly predicts when to begin grasping the block. Additional results are shown in the appendix.

maintaining a track prediction horizon, predicting query point information, and encoding point tracks into features for use in downstream tasks. To train our model, we create a pseudo-labeling pipeline using point trackers and create new benchmarks and metrics to evaluate the track prediction performance of our model. Lastly, we demonstrate that our track prediction model can significantly improve the success rate in robotics and the quality of predicted motions in interaction synthesis.

Though our model is capable of zero-shot track prediction on unseen video datasets, it still has limitations. First, we find that our method inherits the limitations of point tracker models, such as background points being dragged along with foreground moving objects and tracks jumping around on solid colored areas. Second, while our model can accurately model camera motion, track prediction in general is unreliable in scenarios of large camera motion where none of the original points are visible in the frame.

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

| Track generation pre-training data | Split | Tasks completed in a row | | | | | |
|---|---|---|---|---|---|---|---|
| | | 1 | 2 | 3 | 4 | 5 | Avg. Len. |
| Robotics only | ABC→D | 0.913 | 0.808 | 0.719 | 0.636 | 0.558 | 3.60 |
| Robotics + Human motion | ABC→D | **0.917** | **0.816** | **0.723** | **0.641** | **0.561** | **3.66** |
| Robotics only | 10% data | 0.835 | **0.597** | **0.407** | **0.299** | **0.210** | **2.32** |
| Robotics + Human motion | 10% data | **0.838** | 0.592 | **0.407** | 0.293 | 0.209 | **2.32** |

Table 4: CALVIN Benchmark pre-training dataset zero-shot comparison.

## A Pre-training data comparison

In Table 4 we present results where we ablate the presence of human motion data in our pre-training dataset and its performance impact on the CALVIN robotics benchmark. We keep all elements of the model exactly the same with the exception of pre-training dataset mixture. We find that robotics performance is not significantly impacted by the addition of human motion data. While this is not surprising, given that the two domains are significantly far apart, it also suggests that motion prediction can be easily merged into a single foundation model under our framework. In order to observe some level of positive transfer, we would likely need to train on some linking dataset such as HowTo100m Miech et al. (2019), which has data relevant to both domains.

## B Implementation details

For our feature fusion module, the local image features are extracted using a SAMv2 (Ravi et al., 2024) tiny model, text encoding is performed using the language branch of the CLIP (Radford et al., 2021) ViT-B/32 and global image features are extracted using the image branch of the CLIP ViT-B/32. Our feature fusion module has the same size as a ViT-Base (Dosovitskiy et al., 2020), and the flow matching predictor is the same size, but with half the number of layers. All videos are sized to 256 x 256 for training and inference and the model is trained for 500k steps to predict a maximum of 100 points over 50 video timesteps of variable frame rates. We provide more in depth implementation details in the appendices.

The query predictor model also uses the SAMv2 (Ravi et al., 2024) model, and uses a simple MLP on top of the selected SAMv2 features to make a movement prediction. We optimize our track generation models and query prediction models using AdamW (Kingma & Ba, 2014; Loshchilov & Hutter, 2017) with a batch size of 16 and learning rate of $5 \times 10^{-5}$. We use consistency training with an EMA of 0.9999 updated every 100 training steps and sample from the EMA weights during inference. For the track generation model, training is performed on 8x L40S GPUs over the course of five days. The motion generator architecture matches that of a vision transformer Dosovitskiy et al. (2020), except we alternate between spatial attention and causally masked temporal attention. The flow matching predictor is also a vision transformer architecture, but with only spatial attention layers. Our feature fusion module has the same model dimensions as a ViT-Base, and the flow matching predictor has the same dimensions, but with half the number of layers.

During inference, we sample using 16 denoising steps and horizon updates are performed using 4 denoising steps and a timestep of 0.8 (80% previously generated track, 20% noise). Our GR-1 implementation follows that of prior work (Wu et al., 2023), and we train a point track feature extractor of the same size as ViT-Base (Dosovitskiy et al., 2020). For robotics experiments, we use a horizon length of 16 for conditioning updated in an online fashion for the entire episode, and for point prediction results, we predict a horizon of 50 given a single conditioning frame.

In human object interaction synthesis, we use a predicted track horizon of 50 timesteps [1] from a single starting frame and we use a track encoder model with the same model dimensions as a ViT-Base (Dosovitskiy et al., 2020) model.

---

[1]We downsample the original rendered video by a factor of 3, so this translates to approximately 5 seconds.

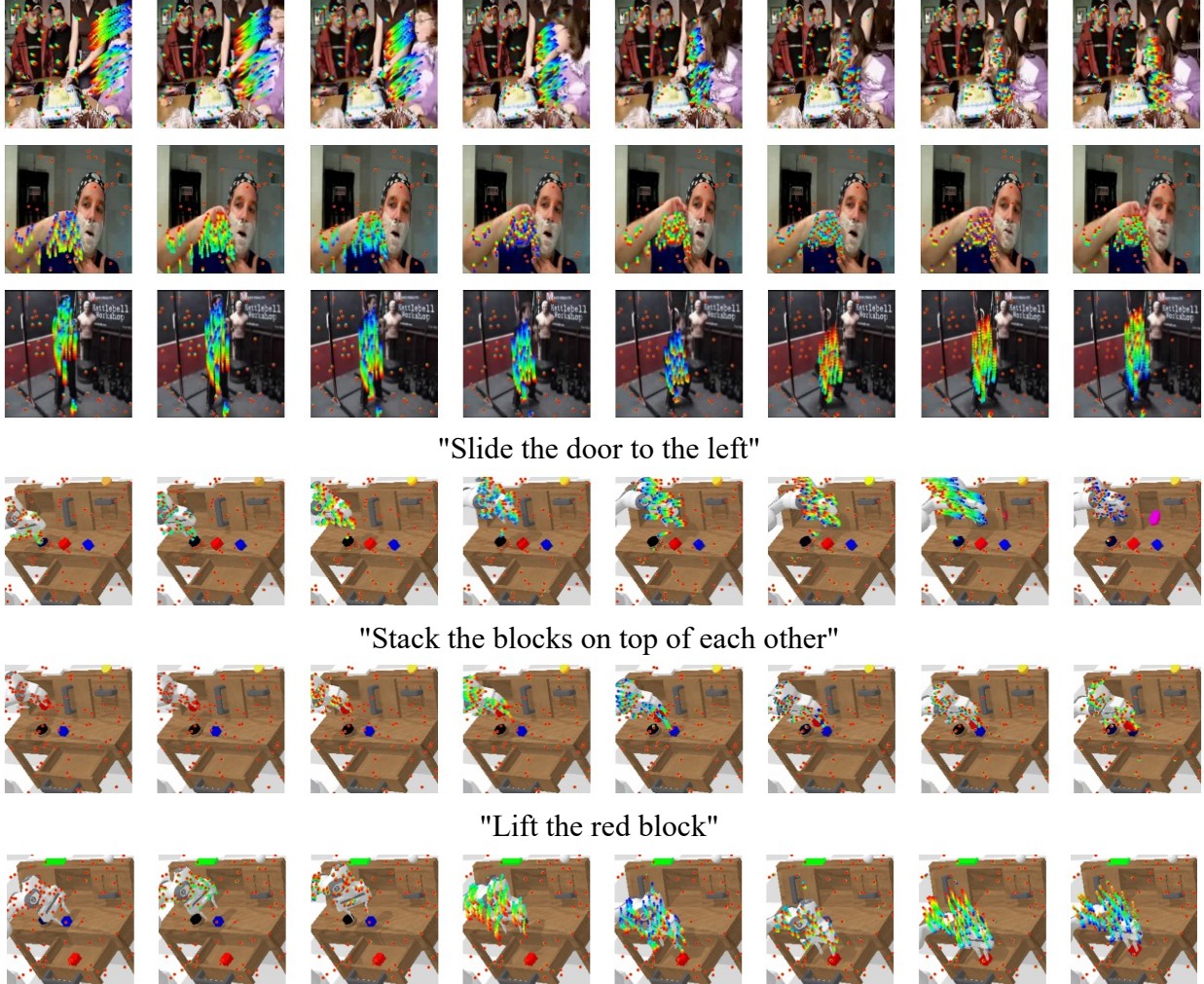

Figure 6: **Motion generation visualizations.** In the top three rows we show results from the CALVIN ABC → D benchmark. In the bottom three rows, we show results for UCF-101 point prediction. Each prediction begins from a blue origin point and transitions to red in 16 timesteps. Note that even though the UCF-101 visualization is generated from a single starting frame 50 frames into the future, we visualize future tracks in the same manner as robotics for consistency.

## C   Qualitative Results

Here, we visualize additional samples of predicted motion from our zero-shot model. Also, we attach video clips of our samples in the supplementary materials. In the very top row, we see our model is capable of interpreting intent as it can predict the person leaning forward to blow out candles then leaning back to their original position. In the second from top example we see a similar example where the model is able to predict the correct shaving motion. In the third from the top example we see the model properly predict someone performing a squat with the proper up and down motion predicted.

In the top CALVIN sample, we observe our model properly guiding the robot arm towards the sliding door handle, and predicts the correct slide direction once arrived. In the middle sample we observe our model

Table 5: Comparison of prediction objective (regression vs flow matching) on UCF-101 and CALVIN ABC → D by $< \delta^x$, measuring the ratio of points within $x$ pixels of the target and $ADE$ measuring the average displacement error between predicted and ground truth tracks at 256×256 resolution.

| Method | UCF-101 | | | | | | | CALVIN ABC → D | | | | | | |
|---|---|---|---|---|---|---|---|---|---|---|---|---|---|---|
| | $< \delta^4$ | $< \delta^8$ | $< \delta^{16}$ | $< \delta^{32}$ | $< \delta^{64}$ | $< \delta^x_{\mathrm{avg}}$ | $ADE$ | $< \delta^4$ | $< \delta^8$ | $< \delta^{16}$ | $< \delta^{32}$ | $< \delta^{64}$ | $< \delta^x_{\mathrm{avg}}$ | $ADE$ |
| regression | 0.237 | 0.370 | 0.540 | 0.742 | 0.911 | 0.560 | 25.8 | 0.362 | 0.397 | 0.465 | 0.591 | 0.823 | 0.528 | 35.5 |
| flow matching | **0.237** | **0.386** | **0.581** | **0.788** | **0.935** | **0.586** | **24.5** | **0.365** | **0.405** | **0.486** | **0.659** | **0.909** | **0.565** | **33.6** |

predicting the correct movement direction and distance necessary to stack one block on top of another. Lastly in the bottom sample we observe our model predicting the correct movement direction toward the red block, then once the robot arm reaches the block location, an upward motion to lift the block.

## D  Avoiding train test time mismatch

In our autoregressive flow matching architecture, we use separate modules for past input conditioning and denoising prediction. Aside from providing inference speed benefits, there is also a train test time mismatch when concatenating noise at the start of the model, which we believe is important to discuss.

To illustrate this problem, consider a simple example with a single-dimensional input of length three $x_1, x_2, x_3$ where the goal is to predict the next continuous data point conditioned on the previous. Specifically, $x_2$ conditioned on $x_1$, and $x_3$ conditioned on $x_1$ and $x_2$. Now, suppose we do not decouple past input conditioning and denoising prediction and instead opt to use a single merged model meaning we concatenate our noised predicting sample at the beginning of a causally masked transformer model. During training time, we sample a denoising timestep $t \sim Uniform(0, 1)$ and for each of the two positions we are predicting noise values $\epsilon_2, \epsilon_3 \sim \mathcal{N}(0, 1)$ which produces noised targets $x_2^t = tx_2 + (1-t)\epsilon_2$ and $x_3^t = tx_3 + (1-t)\epsilon_3$. During training time, to make the prediction the third position, the model observes $x_2^t$ which is $x_2$ noised at timestep $t$ to denoise $x_3^t$ which is also noised at the same timestep $t$. Now consider an inference pass where we now need to perform multiple denoising steps to fully denoise from $\epsilon_3$. In order to match what the model observes at training time, for each intermediate denoising step $t_i$, we must also provide an $x_2$ noised at the same $t_i$. To do this, we would need to store the entire denoising path of $x_2$ for each $t_i$. Since we are using a causally masked transformer model, we would then be required to expand our key-value cache size by a factor equal to the number of denoising steps which in most scenarios is simply infeasible. If we were to not store values, then we cannot provide an $x_2$ noised to the same $t_i$ denoising timestep, and cached keys and values or an $x_2$ noised to that denoising timestep.

One potential counter argument is that the model could learn to not attend to prior noised values given it already has the fully denoised value as input. While this is may be possible by learning a value projection in all attention layers orthogonal to the information of prior noised values in the latent space, one may not want to provide the fully denoised value as input. For example, in our model we do not directly provide the positional information into our feature fusion module as we find this caused overfitting to the point location distribution. Therefore, in order to avoid this train test time mismatch, it is best to separate the modules for past conditioning and denoising prediction such that the denoising prediction does not observe previously noised values while also gaining inference speed benefits.

## E  Prediction objective ablation

While our method uses a flow matching prediction objective to accurately model the uncertainty of point tracking, we could also use a simple squared regression objective. To justify using flow matching, we ablate the prediction objective in Table 5. We find that flow matching significantly outperforms simple squared regression across all metrics and also enables the sampling of a variety of future motion generations.

## F Prediction Latency

One advantage of our method and problem formulation is fast inference speed in comparison to video models. For 100 query points, the forward pass time is approximately 23 ms on a 3090 GPU. Using a flow-matching process with 16 sampling timesteps, this corresponds to approximately 268 ms to produce a sample from scratch and 92 ms for an update step requiring 4 sampling steps. Furthermore, if a faster inference was required we can reduce the number of query points or distill to a 1 step model using various methods developed to distill flow matching models.

