# OpenReview forum: "Autoregressive Flow Matching for Motion Prediction"
_TMLR — Rejected by TMLR_

### Review · Reviewer_oaNK · 2026-04-01

**Summary Of Contributions:**

This paper proposes a novel method, autoregressive flow matching (ARFM), for motion prediction. The method is based on the idea of using a flow-based model to predict the future trajectory of an object given its past trajectory. Their new model has two main components: 1) feature fusion module, a spatiotemporal transformer trained with input features (global image feature, local image features, and text embeddings) coming from three foundational models; 2) flow matching predictor, a spatial attention transformer trained with relative shifts of matching tracking points, some noise and the fused features. The authors suggest to leverage off-the-shelf point tracker to generate training data. The authors demonstrate that their method outperforms existing methods on several benchmark datasets.


## Strengths

- The two-stage split (fusion encoder vs lightweight per-step denoiser) reduces repeated heavy computation (i.e. the fusion features) during inference. This is an important aspect for real-time applications.
- Empirical results show that track prediction improves over diverse baselines and show downstream utility in robotics and interaction synthesis domains. The paper also provide ablation studies (regression vs. flow matching) to show the contribution of each component of the proposed method.

## Weaknesses

- The proposed techniques depends heavily on the quality of the point tracker, which may not be robust in real-world scenarios. Did the authors evaluate the performance and most importantly the variance of the off-the-shelf point tracker on the datasets used in the paper? If the point tracker fails to track the object accurately, it may lead to poor performance of the flow matching predictor. In any case, it might be worth reporting the statiscal variance of the point tracker on the datasets used in the paper, to better understand the potential limitations of the proposed method.

- I am a bit confused about the motivation behind proposing a novel motion prediction method given that in the related work section, the authors says "With these recent advancements, we are able to use existing point trackers to pseudo-label videos for training our track prediction model". What are the limitations of relying on existing point trackers that the proposed method is trying to address? I believe the argument lies in having a single model that can replace the entire pipeline needed for motion prediction. Or maybe it is that the existing point trackers are deterministic and do not capture the uncertainty in the motion prediction task, while the proposed method can model the uncertainty through the flow-based approach. It would be helpful if the authors could clarify this point in the paper.

**Audience:**

Yes

**Audience Explanation:**

Yes. On top of the proposed technique that would be of interest to researchers in motion prediction, the paper present a reusable design pattern for probabilistic sequence modeling of continuous variables where you need autoregressive rollout, multi-modal conditioning, and efficient sampling.

**Broader Impact Concerns:**

While the authors do briefly mention failure modes inherited from point trackers and difficulty with large camera motion when points leave frame, a broader impact statement would help the readers better understand how concretely those issues would affect real-world deployment of this technology.

**Claims And Evidence:**

Yes

**Claims Explanation:**

- For the most part, the claims made in this paper are supported by empirical evidence. Table 1 and 5 show improvement over baselines on track prediction tasks. Table 2 and 3 show usefulness of the proposed method in downstream tasks. Statistical significance could be reported to further support the claims made in the paper.

- There is some claims in the paper about the efficiency of the proposed method compared to existing methods, but it is not clear how the authors measured this efficiency. It would be helpful if the authors could provide more details about the computational resources used for training and inference, as well as the time taken for each method. This would allow readers to better understand the practical implications of the proposed method.

**Requested Changes:**

- In the table 2, as someone outside the field of motion prediction, it is not clear to me what the metrics are exactly. I could find some of them defines in the paper, but not all of them. It would be helpful to have a clear definition of each metric used in the table, either in the main text or in a supplementary material. This would allow readers to better understand the results and compare them with other methods.

- For the experiments reported in table 2 and table 3, it is unclear to me what zero-shot vs. fine-tuned means in the context of this paper. For instance, in the GR-1 + ARFM experiment (finetuned), does zero-shot mean that the GR-1 part of the pipeline was fine-tuned to handle the ARFM features? Or does it mean that the ARFM part of the pipeline was fine-tuned on that particular task and GR-1 remains frozen? Or end-to-end fine-tuning?

### Minor

Several citations are misformatted throughout the paper.

- p.2 in the second contribution point, should it be "human interaction synthesis"?
- p.3 missing words in "...to train on method completely on unlabeled video sources"
- p.5 (last line of the first paragraph of "Flow matching predictor"): missing space before with.
- p.6 Fig 4. "previously predicts tracks" should be "previously predicted tracks"
- p.7 "a singular trained application model" should be "a single trained application model"
- p.9 table 3. Typo. "signficantly" should be "significantly"
- p.14 "we use a predict a track horizon"
- p.8. Table 2, the last row is very pale and hard to read.

---

> ### Author Response · Authors · 2026-05-03
> **Response to Review**
>
> > In the table 2, as someone outside the field of motion prediction, it is not clear to me what the metrics are exactly.
>
> We have modified the paper to clarify that we are using the exact same metrics and settings as the original FullBodyManipulation paper.
>
> > For the experiments reported in table 2 and table 3, it is unclear to me what zero-shot vs. fine-tuned means in the context of this paper.
>
> Thanks for raising this point. In both the zero-shot and fine-tuned cases, the downstream model is tuned to take the point tracks as input (it unfortunately is not feasible to use point tracks as input in another way since the base models did not use point tracks). In the zero-shot setting we take our pre-trained motion predictor which has not seen any data within the calvin benchmark and use that to predict tracks, while in the fine-tuned setting we fine-tuned our pretrained model using pseudo labeled data (the same pseudo-labeling process described in the paper) to fine-tune the track prediction on the domain. The same applies for the CHOIS benchmark. We have modified the paper to clarify this.
>
> > Minor
>
> Thanks for raising these typos. We have modified the paper to fix these
>
> > p.8. Table 2, the last row is very pale and hard to read.
>
> We specifically made this row transparent because it’s an “oracle” method and not really comparable. However, we have now made it less transparent to make it more readable.

---

### Review · Reviewer_staM · 2026-04-01

**Summary Of Contributions:**

The paper proposes a formulation for generalized motion prediction using autoregressive flow matching (ARFM), combining a feature-fusion module with a lightweight flow-matching predictor to model future point tracks efficiently. The paper also goes beyond pure prediction and shows downstream gains in robotics and interaction synthesis, which makes the work feel practically meaningful rather than just benchmark-driven.

strength:
* predicting sparse future point tracks instead of full video is a smart middle ground between domain-specific motion models and expensive video generation.
* The empirical evaluation is broad, covering multiple datasets, domains, and downstream tasks, which helps support the paper’s claim of more generalized motion prediction.

weakness:
The paper is somewhat hard to read in places. While the ideas are interesting, parts of the presentation could be made clearer to improve readability and help readers more easily follow the method and its motivation.

**Audience:**

Yes

**Audience Explanation:**

I think the problem and its applications are interesting, and this is an active research area with broad interest from the community.

**Claims And Evidence:**

Yes

**Claims Explanation:**

Partially to mostly yes. The empirical evidence is reasonably convincing for the paper’s practical contributions, especially the usefulness of the method across diverse datasets and downstream tasks. The results on track prediction, interaction synthesis, and robotics support the claim that the proposed approach is broadly useful. However, the evidence is somewhat less definitive for isolating the specific contribution of ARFM itself, since the main paper does not fully disentangle the impact of the autoregressive flow-matching formulation from other components of the overall system, such as feature fusion, pseudo-labeling, and horizon updating. Thus, the paper supports its overall contribution well, but supports some individual contribution claims more strongly than others.

**Requested Changes:**

critical:
1. Since the method includes several components beyond ARFM itself, such as the feature fusion module, query information predictor, horizon updating strategy, and track encoder, it would significantly strengthen the paper to include ablations showing the contribution of each part. This is particularly important for understanding how much of the performance gain comes from the proposed autoregressive flow matching formulation versus the surrounding system design.
2. Section 4.3 is somewhat under-explained. Table 2 contains many metrics, but the paper does not clearly define them in the main text, making the results hard to interpret for readers unfamiliar with the interaction-synthesis literature. In addition, the discussion of Table 2 lacks intuition about which improvements are most important and why future track conditioning leads to those gains.
3. The presentation of the experimental results could be improved. In both Table 2 and Table 3, several reported metrics are not clearly defined in the main text, and the discussion provides limited intuition for how to interpret them.
4. The paper models motion through frame-to-frame relative shifts, but it is unclear why this parameterization is preferred over alternatives such as absolute displacement from the first frame. I did not see experimental evidence or discussion justifying this choice, and a comparison would strengthen the paper.
5. One of the central methodological points of the paper is the separation between conditioning on past inputs and the denoising procedure, but this distinction is currently not as clear as it could be. A more explicit explanation, together with a simple schematic figure, would significantly improve the readability of the method section.

Non critical:
1. It isn't clear the zero-shot meaning here in the paper, I would suggest to give some explanation regarding that.
2. The paper could better clarify the autoregressive aspect of the method. As I understood it, flow matching is applied at each timestep to predict the current track, and this prediction is then fed back, together with past context, to condition the next timestep. If so, the autoregressive behavior arises from the sequential rollout, while flow matching is the per-step prediction mechanism. Making this distinction clearer would improve the presentation
3. "To subsample these tracks, we first perform a minimum visibility ratio filtering where
only tracks which are visible for greater than half of the future frames are selected for prediction." this is a little confusing, I would suggest give some intuition of why this is happening
4. "Additionally, for modeling we transform the tracks such that the input tracks are every track except for the
last video timestep TI = T[:, :-1] and the prediction targets TT = T[:, 1:] − T[:, :-1] are the relative shift for
the next track." I think this sentence in 3.1 needs a little refinement

---

> ### Author Response · Authors · 2026-05-03
> **Response to review (1/2)**
>
> > Since the method includes several components beyond ARFM itself, such as the feature fusion module, query information predictor, horizon updating strategy, and track encoder, it would significantly strengthen the paper to include ablations showing the contribution of each part.
>
> The reason we don’t specifically ablate these components is they are necessary components to perform some task and not just for an empirical performance improvement. While there are certainly many possible ways to design these modules, we are creating an initial method for each of these tasks, and we do not claim we have created the best possible version of each of these modules. The feature fusion module is necessary to capture video and track information to make an accurate prediction. The query information predictor only exists because many downstream applications do not readily provide a query point set. While a user could specify their own query points we would like for the entire process to work automatically end to end. The horizon updating strategy is only needed for robotics for efficient inference and maintaining temporal consistency rather than independently forecasting each new frame. The track encoder is needed as an interface between the track predictions and the downstream application.
>
> > Section 4.3 is somewhat under-explained. Table 2 contains many metrics, but the paper does not clearly define them in the main text, making the results hard to interpret for readers unfamiliar with the interaction-synthesis literature. In addition, the discussion of Table 2 lacks intuition about which improvements are most important and why future track conditioning leads to those gains.
>
> Thank you for raising this point, we should have clarified that we use the exact same metrics and settings as the FullBodyManipulation benchmark paper. We have modified the paper to clarify this.
>
> > The presentation of the experimental results could be improved. In both Table 2 and Table 3, several reported metrics are not clearly defined in the main text, and the discussion provides limited intuition for how to interpret them.
>
> Similar to the above issue we have now clarified that we use the same metric calculation and will refer the reader to the corresponding benchmark as it is common practice to not re-define metrics if they are the same as an existing benchmark.
>
> > The paper models motion through frame-to-frame relative shifts, but it is unclear why this parameterization is preferred over alternatives such as absolute displacement from the first frame. I did not see experimental evidence or discussion justifying this choice, and a comparison would strengthen the paper.
>
> We initially tried absolute displacement, however we found the training to be unstable and the loss diverged. When using the frame to frame relative shift formulation we found the training to be much more stable. Since the training diverged the predictions were generally random which is why we did not report this result.
>
> > One of the central methodological points of the paper is the separation between conditioning on past inputs and the denoising procedure, but this distinction is currently not as clear as it could be. A more explicit explanation, together with a simple schematic figure, would significantly improve the readability of the method section.
>
> The key idea is that the past conditioning is only done in the feature fusion module which outputs features which have already taken into account the past track positions and visual features. Then the flow matching predictor operates only on the current frame and the past conditioned features. This enables us to denoise online quickly by only needing to denoise one timestep at every inference pass and avoids a train-test time mismatch with respect to the denoising prediction seeing partially noised past samples at train time and only fully denoised samples at test time. We discuss this concept in greater detail in Appendix D, but we will modify the paper to make this more clear.
>
> > It isn't clear the zero-shot meaning here in the paper, I would suggest to give some explanation regarding that.
>
> By zero-shot we mean we are testing on unseen datasets with distribution shift compared to our training data, which we believe is a common definition.

---

> ### Author Response · Authors · 2026-05-03
> **Response to review (2/2)**
>
> > The paper could better clarify the autoregressive aspect of the method. As I understood it, flow matching is applied at each timestep to predict the current track, and this prediction is then fed back, together with past context, to condition the next timestep. If so, the autoregressive behavior arises from the sequential rollout, while flow matching is the per-step prediction mechanism. Making this distinction clearer would improve the presentation
>
> Yes this is why it is autoregressive. We have modified the paper to clarify this.
>
> > "To subsample these tracks, we first perform a minimum visibility ratio filtering where only tracks which are visible for greater than half of the future frames are selected for prediction." this is a little confusing, I would suggest give some intuition of why this is happening
>
> Thank you for raising this point. We specifically do this to increase the amount of supervision per training point (we only supervise on timesteps where the track is visible). We have modified the paper to clarify this.
>
> > I think this sentence in 3.1 needs a little refinement
>
> We have edited the paper to clarify this.

---

### Review · Reviewer_vua1 · 2026-04-03

**Summary Of Contributions:**

The paper proposes an autoregressive flow matching approach to predict future point track location in motion videos. The approach uses pseudo-labels from off-the-shelf point trackers to train on video datasets, predicting future point track locations over long horizons with vision-language conditioning. The training pipeline uses BootsTAP point tracker to generate pseudo-labels from unlabeled videos, with track filtering based on visibility and movement-weighted sampling.
The approach separates past conditioning from denoising prediction, avoiding train-test mismatch and enabling efficient online updates
The proposed mnechanism predicts only sparse point tracks rather than full video frames, reducing computational cost

I have the folloiwing concerns:
- Limited Evaluation: Uses only 2 benchmarks (CALVIN and UCF-101 point prediction), insufficient to validate generality claims
- No Zero-Shot Testing: Doesn't demonstrate generalization to unseen domains despite claiming "zero-shot predictions"
- Missing Comparisons: No comparison against video generation models to support claims about motion accuracy and computational efficiency
- Computational Cost: Still requires significant resources (8x L40S GPUs for 5 days)

**Audience:**

Yes

**Audience Explanation:**

the topic is interesting and the approach has merits

**Claims And Evidence:**

No

**Claims Explanation:**

the evaluation is not sufficient to support generalisation, lack of comparative analysis

**Requested Changes:**

-  Generalization Claims: The paper claims zero-shot generalization but only tests on domains present in training data. How would the model perform on truly unseen domains?
-  Video Generation Comparison: How does ARFM compare to recent video generation models in terms of motion accuracy and computational cost? The paper claims these models "struggle to accurately model complex motions" but provides no evidence.
-  Track Quality: What impact does the quality of the pseudo-labels from BootsTAP have on prediction accuracy? Are there failure modes where poor tracking degrades performance?
- Real-time Performance: The paper mentions efficient inference but doesn't provide latency numbers.

---

> ### Author Response · Authors · 2026-05-03
> **Response to review**
>
> > Generalization Claims: The paper claims zero-shot generalization but only tests on domains present in training data. How would the model perform on truly unseen domains?
>
> We believe that CALVIN would be considered “zero-shot” considering we haven’t trained on any robotics samples within their environment. We considered the method “zero-shot” in the sense that we are training on entirely different datasets than the ones we are evaluating on which have their own distribution shifts. Similarly we would consider video generators such as iVideoGPT and CosmosVideo2World as “zero-shot” even though they train very heavily on the domains we are evaluating them on.
>
> > Video Generation Comparison: How does ARFM compare to recent video generation models in terms of motion accuracy and computational cost? The paper claims these models "struggle to accurately model complex motions" but provides no evidence.
>
> We compare to video generation models, specifically Cosmos Video2World and iVideoGPT, in our point prediction benchmarks (Table 1) and outperform them across all metrics. Generally we find the motion modeling of these video models to perform quite poorly despite being trained on a large amount of data.
>
> > Track Quality: What impact does the quality of the pseudo-labels from BootsTAP have on prediction accuracy? Are there failure modes where poor tracking degrades performance?
>
> We discuss the limitations of the pseudo labels in our main paper. Specifically we find that there are some failures such as nearby background points being dragged with foreground objects, and tracks occasionally jumping around on solid colored objects (generally only present in simulators). Our motion prediction does inherit these issues, but generally the trackers produce high quality labels and our method allows us to easily swap to new ground truth labels as trackers improve.
>
> > Real-time Performance: The paper mentions efficient inference but doesn't provide latency numbers.
>
> For 100 query points, the forward pass time is approximately 23 ms on a 3090 GPU. Using a flow-matching process with 16 sampling timesteps, this corresponds to approximately 268 ms to produce a sample from scratch and 92 ms for an update step requiring 4 sampling steps. Furthermore, if a faster inference was required we can reduce the number of query points or distill to a 1 step model using various methods developed to distill flow matching models. We have revised the paper to add these details.

---

### Decision · Action_Editor_vEPd · 2026-06-18

**Recommendation:** Reject

**Audience:**

Yes

**Audience Explanation:**

The paper is of general interest to researchers working on motion prediction from video.

**Claims And Evidence:**

No

**Claims Explanation:**

The paper presents an autoregressive flow matching (ARFM) approach for generalized motion prediction of future track locations in video.  The reviewers agreed that the empirical results are promising and that separating feature fusion from flow prediction is an interesting and practical design. However, multiple reviews raised concerns about the strength of the generalization claims, the lack of ablations to determine the contribution of ARFM from other system components, various concerns on the evaluation methodology, computational cost, and the dependence of the method on the quality of the trackers. The rebuttal clarified some aspects of the concerns, especially regarding zero-shot evaluation, computational performance (which remains expensive), and some details on method and presentation. However, after discussion and considering the rebuttal, while the paper shows promise, the reviewers contend that the evidence does not fully support the paper’s broader claims, and that stronger empirical validation and analysis, especially ablations, are needed before this paper is ready for publication.

**Resubmission Of Major Revision:**

The authors may consider submitting a major revision at a later time.